# Hypoxia Triggers TAZ Phosphorylation in Basal A Triple Negative Breast Cancer Cells

**DOI:** 10.3390/ijms231710119

**Published:** 2022-09-04

**Authors:** Qiuyu Liu, Wanda van der Stel, Vera E. van der Noord, Hanneke Leegwater, Bircan Coban, Kim Elbertse, Joannes T. M. Pruijs, Olivier J. M. Béquignon, Gerard van Westen, Sylvia E. Le Dévédec, Erik H. J. Danen

**Affiliations:** Leiden Academic Centre for Drug Research, Leiden University, 2333 CC Leiden, The Netherlands

**Keywords:** hypoxia, breast cancer, TAZ, phosphorylation, basal A triple negative cells

## Abstract

Hypoxia and HIF signaling drive cancer progression and therapy resistance and have been demonstrated in breast cancer. To what extent breast cancer subtypes differ in their response to hypoxia has not been resolved. Here, we show that hypoxia similarly triggers HIF1 stabilization in luminal and basal A triple negative breast cancer cells and we use high throughput targeted RNA sequencing to analyze its effects on gene expression in these subtypes. We focus on regulation of YAP/TAZ/TEAD targets and find overlapping as well as distinct target genes being modulated in luminal and basal A cells under hypoxia. We reveal a HIF1 mediated, basal A specific response to hypoxia by which TAZ, but not YAP, is phosphorylated at Ser89. While total YAP/TAZ localization is not affected by hypoxia, hypoxia drives a shift of [p-TAZ(Ser89)/p-YAP(Ser127)] from the nucleus to the cytoplasm in basal A but not luminal breast cancer cells. Cell fractionation and YAP knock-out experiments confirm cytoplasmic sequestration of TAZ(Ser89) in hypoxic basal A cells. Pharmacological and genetic interference experiments identify c-Src and CDK3 as kinases involved in such phosphorylation of TAZ at Ser89 in hypoxic basal A cells. Hypoxia attenuates growth of basal A cells and the effect of verteporfin, a disruptor of YAP/TAZ-TEAD–mediated transcription, is diminished under those conditions, while expression of a TAZ-S89A mutant does not confer basal A cells with a growth advantage under hypoxic conditions, indicating that other hypoxia regulated pathways suppressing cell growth are dominant.

## 1. Introduction

Breast cancer represents a heterogeneous disease with several intrinsic molecular subtypes. A major distinction can be made by referring to the cell type of origin, with HER2-enriched, luminal A, and luminal B breast cancers arising from the luminal epithelial compartment and basal-like breast cancers arising from the basal/myoepithelial cell compartment [1]. Luminal A has a relatively good prognosis and luminal B breast cancer has higher Ki-67 expression and chromosomal instability and is associated with a less favorable prognosis [2,3,4,5]. Basal-like breast cancers can be subdivided into basal A and B classes, based on epithelial (basal A) or more mesenchymal characteristics (basal B) including differences in expression of cell adhesion receptors such as E-cadherin and claudins and most basal-like breast cancers are triple negative breast cancer (TNBC): they lack estrogen receptor and progesterone receptor and do not overexpress HER2. Triple negative breast cancer is the most aggressive subtype, with frequent metastasis, and for which targeted therapies are lacking [6,7,8,9].

In addition to genetic and epigenetic alterations driving initiation and progression of cancer, cross talk between tumor cells and their tissue microenvironment controls primary tumor growth, tissue invasion, and metastatic colonization [10,11]. One aspect of the microenvironment surrounding cancer cells in solid tumors is its hypoxic nature [12,13]. While hypoxia initially poses a restriction to tumor growth, tumors ultimately escape from this barrier and there is clinical evidence pointing to hypoxia as a driver of tumor progression and therapy resistance [14,15,16]. Tumor cells as well as various cell types in the tumor microenvironment adapt gene expression patterns in response to hypoxia, predominantly through the hypoxia-inducible factor (HIF) family of transcription factors. Central in this response is the stabilization of a transcription complex consisting of HIF1α or HIF2α and aryl hydrocarbon nuclear translocator (ARNT; a.k.a. HIF1b). HIF1/2α is ubiquitinated and targeted for degradation by the Von Hippel–Lindau (VHL) E3 ubiquitin ligase complex under normoxia. Under hypoxia, degradation domains in HIF1/2α are not hydroxylated by oxygen-dependent prolyl hydroxylases, thus preventing recognition by VHL and causing stabilization of HIF1/2α and its concentration in the nucleus, formation of the HIF/ARNT complex, and transcriptional activation of HIF target genes [17,18]. In addition, hypoxia-independent mechanisms have been identified that can trigger HIF signaling in normal and cancer cells [12].

Tumor hypoxia has been extensively demonstrated in breast cancer [19] and expression of HIF1α represents an independent factor for poor prognosis in patients with lymph node-negative and -positive breast cancer, including several different subtypes [20,21,22,23,24]. Hypoxia impacts on a variety of other signaling pathways downstream of, or in parallel to, HIF signaling. It has been shown that hypoxia can affect expression, localization or activity of Yes-associated protein 1 (YAP) and WW-domain-containing transcription regulator 1 (WWTR1; a.k.a. TAZ). YAP and TAZ are transcriptional co-activators that regulate gene transcription in complex with members of the TEA domain (TEAD) family [25]. YAP/TAZ activity is regulated at the level of gene expression, protein degradation, and nuclear/cytoplasmic distribution. One critical event is phosphorylation of YAP and TAZ, which shifts the balance towards cytoplasmic localization where interactions with 14-3-3 proteins target them to proteasomal degradation. Multiple inputs, including activity of LATS1/2 kinases in the Hippo signaling cascade, cell adhesion and polarity, extracellular forces, cell metabolism and growth factors control YAP/TAZ phosphorylation, nuclear localization, and interaction with TEADs to activate transcription [25,26]. Reduced YAP/TAZ nuclear localization in response to long term inhibition of prolyl hydroxylase domain enzymes in renal tubular cells was proposed to involve HIF signaling [27]. Increased phosphorylation of TAZ under hypoxia was reported to occur independent of LATS1 in ovarian cancer cells [28]. In breast cancer cells, HIF1 was reported to regulate expression and localization of TAZ and an interaction between-and reciprocal activation of-HIF1 and TAZ was shown [29,30].

It is currently not known to what extent signaling responses to hypoxia differ between different breast cancer subtypes. Here, we investigate the response to hypoxia in a series of luminal and basal A breast cancer cell lines. We find that HIF1α stabilization is similarly activated and triggers overlapping as well as distinct changes in gene expression including YAP/TAZ/TEAD target genes in these subtypes. Strikingly, HIF1 mediated phosphorylation of TAZ, but not YAP, at a site known to promote its cytoplasmic sequestration and proteasomal degradation, is observed in all basal A cell lines tested but not in any of the luminal breast cancer cell lines. Such basal A-specific phosphorylation does not involve activation of the Hippo signaling cascade. Instead, we identify Src and CDK3 as kinases involved in phosphorylation of TAZ in basal A cells under hypoxia and we explore the impact on TAZ localization.

## 2. Results

### 2.1. Shared and Distinct Responses to Hypoxia in Luminal and Basal A Breast Cancer Cells

To determine how luminal breast cancer and basal A cells respond to hypoxia, three luminal and three basal-A type breast cancer cell lines were cultured for 1, 3 and 5 days, respectively, under normoxic (21% O_2_) or hypoxic (1% O_2_) conditions. Hypoxia suppressed growth of MCF7 and had no effect on growth of the other two luminal cell lines (Figure 1A). By contrast, growth of all tested basal A cells was markedly attenuated under hypoxia. To explore activation of HIF signaling in these cells, HIF1α stabilization and transcriptional activation of HIF target genes was analyzed. HIF1α stabilization and nuclear localization were similarly induced after 1 and 3 days under hypoxia throughout the panel of luminal and basal A cell lines (Figure 1B and Appendix A). Likewise, protein expression of the hypoxic biomarker CA9, a HIF1 regulated gene encoding a carbonic anhydrase isoenzyme was induced at 3 and 5 days in hypoxia in all cell lines tested (Figure 1C). Next, TempO-Seq, a high-throughput targeted sequencing technology was used to determine genome wide changes in RNA expression in a luminal and basal A cell line in response to 5 days growth in hypoxia. Using a cutoff of [|Log_2_foldchange| > 1; padj < 0.05], ~2000 versus ~5000 hypoxia responsive genes were identified in MCF7 and HCC1143, respectively (Appendix A). This set contained a series of previously published HIF1 responsive genes [31] that partly overlapped between the two cell lines, including a shared strong upregulation of *CA9* (Figure 1D). GSEA and Metascape identified shared hypoxia-induced changes including those in pathways regulating cell cycle progression and extracellular matrix turnover in luminal and basal A cells (Appendix A). Notably, while the response to hypoxia was the most enriched term identified by Metascape in MCF7 (Appendix A) it was not among the 20 most enriched terms in HCC1143 (Appendix A). Rather, in HCC1143 terms associated with adhesion and migration were prominent. Together, these data show that hypoxia triggers HIF1 stabilization in luminal and basal A cells and causes overlapping as well as distinct changes in gene expression.

### 2.2. Expression of YAP/TAZ/TEAD Complex in Breast Cancer Subtypes and Response to Hypoxia

Cross talk between hypoxia and YAP/TAZ signaling has been reported [32]. Analyzing RNA-Seq data for a panel of 52 human breast cancer cell lines [33], we observed a significant increase in *YAP* and especially *WWTR1* (encoding TAZ) expression in basal A cell lines as compared to luminal cell lines (Figure 2A). Accordingly, by exploring RNA-Seq data from breast cancer patients [34,35], we show that expression of *YAP* and especially *WWTR1* was increased in TNBC tumors as compared to ER positive tumors (Figure 2B). The TEAD transcription factors that associate with YAP/TAZ transcriptional coactivators in the Hippo pathway did not differ between TNBC and luminal cell lines but TEAD3 was significantly lower in basal-B TNBC cell lines as compared to basal-A and luminal cell lines (Appendix A). However, this association was not corroborated in clinical samples where, in fact, TEAD2–4 were all increased in TNBC tumors as compared to ER positive tumors (Appendix A). A positive correlation of expression of *YAP* and *WWTR1* with expression of *HIF1A* was detected in breast cancer patients, with particularly high expression of all three genes in basal-like tumors (Figure 2C). The analysis of our TempO-Seq data did not detect significant hypoxia-induced changes in RNA expression of YAP/TAZ/TEAD complex members except for an increase in TEAD3 in HCC1143 cells (Figure 2D). Moreover, no hypoxia-induced changes in RNA expression of canonical upstream regulators of this transcriptional complex in the Hippo pathway were observed, except for an increase in MST1 in MCF7 cells (Figure 2D).

### 2.3. Shared and Distinct Changes in Expression of YAP/TAZ/TEAD Target Genes in Response to Hypoxia in Luminal and Basal A Breast Cancer Cells

The response of three known YAP/TAZ target genes (CYR61, AXL, CTGF) to hypoxia in two luminal and two basal A TNBC cell lines showed an overall trend of increased expression in hypoxia, with CTGF being significantly enhanced in basal A but not luminal cells (Figure 2E). We next scrutinized hypoxia-regulated genes in MCF7 and HCC1143 cells identified by TempO-Seq for the presence of previously published YAP/TAZ target genes [36,37] (Figure 3A; Appendix A). Most candidate targets found in MCF7 were also detected in HCC1143 (indicated in green). As an alternative approach, we analyzed hypoxia-regulated genes for enrichment of TEAD1–4 binding using DoRothEA [38] (Figure 3B; Appendix A). Approximately half of the candidate TEAD binding genes found in MCF7 by this method were also detected in HCC1143 (indicated in blue). For both methods, the larger overall number of hypoxia-responsive genes identified in HCC1143 as compared to MCF7 (Appendix A) was accompanied by a larger number of hypoxia-regulated candidate YAP/TAZ targets (Figure 3A,B). A small number of genes was identified by both methods, i.e., representing TEAD-binding, previously published YAP/TAZ target genes (Figure 3C). Of these, only *WWC2*, a WWC scaffolding protein involved in Hippo signaling [39] and *ATAD2*, an AAA+ ATPase and bromodomain family member with an as yet poorly understood oncogenic function [40], were shared between MCF7 and HCC1143 (Figure 3C). These findings show that, while modulation of genes encoding elements of the Hippo pathway was not evident, overlapping as well as distinct sets of YAP/TAZ/TEAD target genes were modulated in luminal and basal A cells.

### 2.4. HIF1 Mediated TAZ Phosphorylation at Ser89 in Basal A but Not Luminal Breast Cancer Cells

YAP/TAZ/TEAD target genes may be modulated by hypoxia due to post-transcriptional regulation of YAP/TAZ in response to hypoxia. We therefore analyzed expression and phosphorylation of YAP/TAZ. In addition to specific YAP and TAZ antibodies, an antibody recognizing both p–YAP (Ser127) and p-TAZ (Ser89) was used, and signals were distinguished based on molecular weight. Total YAP and TAZ protein levels were not affected by hypoxia and phosphorylation of YAP at Ser127 in most of the cell lines, an event that is associated with cytoplasmic localization, 14-3-3 binding, and proteasomal degradation [32,41], showed no changes under hypoxia in any of the cell lines (Figure 4A). However, 3- and 5-days culture under hypoxia triggered a striking phosphorylation of TAZ at Ser89, similarly associated with cytoplasmic localization, 14-3-3 binding, and proteasomal degradation [42,43], in each of the basal A cell lines (Figure 4A). By contrast, none of the luminal breast cancer cells showed this response, with MCF7 displaying an opposite pattern with reduced p-TAZ (Ser89) under hypoxia. The basal A specific phosphorylation of TAZ at Ser89 under hypoxia was maintained for up to at least 10 days (Figure 4B). Moreover, the prominent difference between basal A and luminal cells with respect to hypoxia induced phosphorylation of TAZ at Ser89 was unaffected by varying seeding densities, excluding differences in cell-cell contact area that are known to modulate YAP/TAZ [44,45,46] as the underlying mechanism (Figure 4C). p-TAZ (Ser89) showed a slightly lower molecular weight in the Western blot than total TAZ (Figure 4D). To confirm specificity of the p-TAZ (Ser89) signal under hypoxia, *WWTR1* silencing in HCC1143 using SMARTpool siRNA strongly reduced the p-TAZ (Ser89) signal as predicted while control luciferase and GAPDH siRNAs did not (Figure 4E). Phosphorylation of TAZ was also induced by treatment of HCC1143 and HCC1806 basal A cells with the HIF1α stabilizing compound DMOG under normoxic conditions, pointing to a HIF1 mediated response (Figure 4F). Treatment with Verteporfin, causing degradation of TAZ, also led to a corresponding loss of the p-TAZ (Ser89) signal induced by either hypoxia or DMOG, further confirming specificity. In these experiments, treatment with the 14-3-3 inhibitor BV02 did not affect total TAZ or p-TAZ (Ser89) levels in basal A cells under normoxic, hypoxic, or DMOG conditions. Together, these experiments point to a HIF1 mediated, basal A specific response to hypoxia by which TAZ, but not YAP, is phosphorylated at a site known to promote its cytoplasmic sequestration and proteasomal degradation [28,47] and which would consequently prevent TAZ from co-activating transcription of downstream target genes.

### 2.5. Expression of a TAZ-S89A Mutant Does Not Prevent Growth Reduction in Basal A Cells under Hypoxia

We wondered whether the increased TAZ phosphorylation under hypoxia was involved in the decreased growth of basal A cells under these conditions. Under hypoxia, growth of basal A cells was reduced and sensitivity to verteporfin, a disruptor of YAP/TAZ-TEAD–mediated transcription decreased (Figure 5A). To assess if increased expression of unphosphorylated TAZ might confer basal A cells with a growth advantage under hypoxic conditions HCC1806 cells were transduced with a TAZ-S89A mutant. The expression of TAZ-S89A increased the total TAZ level, which, as expected, was not accompanied by enhanced levels of p-TAZ (Ser89) in normoxia or hypoxia (Figure 5B). However, the increased presence of non-phosphorylated TAZ did not confer basal A cells with a growth advantage under hypoxic conditions, indicating that other hypoxia/HIF regulated pathways suppressing cell growth were dominant (Figure 5C).

### 2.6. Basal A Specific TAZ (Ser89) Phosphorylation Is Accompanied by Cytoplasmic Localization in Hypoxia

We next analyzed subcellular localization of YAP, TAZ, and phosphorylated YAP/TAZ in luminal and basal A cells under normoxia or hypoxia. TAZ was mostly concentrated in the nuclei of luminal as well as basal A cells and its localization appeared unaffected by hypoxia (Figure 6A–C). Likewise, YAP localization appeared largely nuclear in luminal and basal A cells and was unaffected by hypoxia (Appendix A). Localization of [p-TAZ(Ser89)/ p-YAP(Ser127)] varied between luminal cell lines, appearing largely excluded from nuclei in MCF7 while being present in nuclei in T47D and BT474. This pattern was not affected under hypoxia. By contrast, localization of [p-TAZ(Ser89)/ p-YAP(Ser127)] under normoxia appeared largely nuclear in basal A cells (with some cytoplasmic signal in SUM149PT) and this switched to significantly more cytoplasmic staining under hypoxia in all basal A lines (Figure 6A,B). To specifically visualize localization of the increased levels of p-TAZ (Ser89) under hypoxic conditions in basal A cells, YAP knockout (KO) HCC1806 cells were generated. In the bulk KO population, most of the YAP signal disappeared (Figure 7A) and a corresponding increased nuclear concentration of TAZ was observed (Figure 7B). Under this condition most of the [p-TAZ(Ser89)/ p-YAP(Ser127)] signal disappeared indicating that the signal detected in WT cells was largely due to p-YAP (Ser127) rather than p-TAZ (Ser89) (Figure 7C). Further analysis of subcellular localization by cell fractionation confirmed an increase in nuclear total TAZ upon YAP KO and clearly showed that p-TAZ (Ser89) under hypoxic conditions in HCC1806 was largely localized in the cytoplasm (Figure 7D,E).

### 2.7. Involvement of c-Src and CDK3 in TAZ Ser89 Phosphorylation under Hypoxia

We addressed which kinase could be responsible for phosphorylation of TAZ under hypoxia in basal A cells. No marked alterations in expression of kinases in the canonical upstream module of the Hippo pathway were identified in basal A under hypoxia (Figure 2D) and silencing of MST1, LATS1, or LATS2 did not affect hypoxia induced TAZ Ser89 phosphorylation in HCC1143 cells (Figure 8A). Src activity has been implicated in the upstream regulation of YAP/TAZ/TEAD transcriptional activity in different biological settings [48,49,50,51,52,53,54,55,56]. We observed a marked inhibition of hypoxia induced p-TAZ (Ser89) upon exposure to the Src inhibitor PP2 in three basal A cell lines, but p-Src (Y416) was only weakly affected (Figure 8B). By contrast, PP2 did not inhibit p-TAZ (Ser89) in luminal cells under normoxic or hypoxic conditions (Figure 8C). Another Src inhibitor, Dasatinib also reduced p-TAZ (Ser89) in HCC1143 basal A cells under hypoxia and this compound did reduce p-Src (Y416) levels as expected (Figure 8D). Together, these results suggested that the tyrosine kinase Src was involved in hypoxia induced TAZ (Ser89) phosphorylation in basal A cells but also pointed to an off-target effect of PP2 based on its weak effect on p-Src (Y416) levels. From a series of ChEMBL predicted PP2 targets, the serine/threonine-protein kinase CDK3 stood out in the top 10 as it scored much lower for predicted activity to interact with Dasatinib (Figure 8E; Appendix A). Silencing CDK3 in HCC1143 cells using an siRNA SMARTpool as well as expression of CDK3 shRNAs, significantly reduced p-TAZ (Ser89) in these basal A cells under hypoxia (Figure 8F,G). Together, these results indicated that hypoxia/HIF1 mediated TAZ Ser89 phosphorylation is supported by the activity of c-Src and involves an off target PP2 substrate, identified as CDK3.

## 3. Discussion

In this work, we explore hypoxia regulated changes in gene expression in luminal breast cancer versus basal A TNBC cells and focus on YAP/TAZ signaling. We show that HIF stabilization and expression of a key HIF target, CA9 are similarly induced in response to hypoxia in both breast cancer subtypes. Despite conservation of the canonical HIF-mediated response to hypoxia, the impact of hypoxia on genome wide gene expression differs between luminal and basal A cells. The response to hypoxia for YAP/TAZ target genes, identified based on existing literature as well as predicted to harbor TEAD binding sites is limited. Notably, prediction of enhanced YAP/TAZ activity based on changes in target gene expression is hampered by the fact that other transcription factors are involved as well, including for instance cross talk between TAP/TAZ-TEAD and MRTF–SRF transcriptional complexes in gene regulation [25,57].

Our analyses showing that TAZ, and to a lesser extent YAP RNA levels are higher in basal-like breast cancer cell lines and in TNBC patients as compared to luminal breast cancer cell lines and ER positive patients, respectively agree with earlier work. TAZ protein expression has been reported to increase from barely detectable in low-grade invasive ductal breast carcinomas to strong nuclear staining in ~80% of high-grade invasive ductal breast carcinomas [58]. Studies have associated TAZ expression levels with the TNBC subtype and amplification of the *WWTR* gene can explain only a fraction of the overexpression cases indicating that other mechanisms must be at play (reviewed in [59]). TAZ may be involved in the increased aggressiveness of TNBC by promoting breast cancer stem cell self-renewal and tumor initiation capacity [58,59].

We reveal a HIF1 mediated response to hypoxia occurring in all basal A TNBC cells tested, but not in any of the luminal breast cancer lines, which leads to phosphorylation of TAZ at a site known to suppress nuclear localization and transcriptional activity. This striking response is not affected by cell density, which has a major influence of YAP/TAZ activity [44,45,46]. Similar findings have been reported for ovarian breast cancer cells, indicating that this response may be conserved among distinct cancer types and, in agreement with our study, the Hippo pathway kinase Lats1 was not involved [28]. Our results do not corroborate an earlier report showing that HIF-1 transcriptionally activates the WWTR gene in breast cancer cells [29,30]. No increase in total TAZ levels nor TAZ nuclear accumulation is observed by us upon switching any of the cell lines from normoxia to hypoxia.

We show that hypoxia-induced phosphorylation of TAZ at Ser89 in basal A TNBC cells is not mediated by the canonical Hippo kinases [25,26]. Rather, our results indicate that c-Src supports this event and an off target of the PP2 Src inhibitor, CDK3 is involved. Src family kinases have been previously implicated in phosphorylation of YAP but in that case considered as positive regulators of YAP activity [60,61]. We find that the phosphorylation of TAZ at Ser89 in hypoxic basal A TNBC cells is accompanied by an increase in cytoplasmic localization suggesting that in this case activity of TAZ is attenuated. Interestingly, recent work by others has shown that CDK7 can phosphorylate YAP, but in that case, it involves an activating phosphorylation at Ser128 (adjacent to the inactivating phosphorylation site, Ser127) that protects YAP from ubiquitination [62]. Interactions of CDKs with TAZ have not been previously described. Notably, the dominance of YAP phosphorylation (revealed in our experiments by the major loss of [p-TAZ(Ser89)/ p-YAP(Ser127)] upon deletion of YAP), likely masks functional consequences of TAZ Ser89 phosphorylation for cell survival and proliferation in basal A TNBC cells under hypoxia.

In conclusion, we find that even though HIF1α stabilization is activated similarly in basal and luminal breast cancer cells the response to hypoxia differs with distinct patterns of gene regulation. HIF1 mediated phosphorylation of TAZ, but not YAP, at a site known to promote its cytoplasmic sequestration and proteasomal degradation, is observed in basal A but not luminal cells. We show that Src and CDK3 are implicated in this response. Further studies investigating the mechanism and functional consequence of such modulation of TAZ are warranted.

## 4. Materials and Methods

### 4.1. Cell Culture, Reagents, and Antibodies

Human breast cancer cell lines representing luminal-like (MCF7, T47D, BT474) and basal A (HCC1806, HCC1143 and SUM149PT) subtypes were obtained from the American Type Culture Collection. Cells were cultured in RPMI1640 medium with 10% fetal bovine serum, 25 U/mL penicillin and 25 μg/mL streptomycin in the incubator (37 °C, 5% CO_2_). For normoxia 21% O_2_ was used and for hypoxia 1% O_2_ was used. For experiments, initial seeding cell densities for different cell lines were adjusted to ensure the final cell confluency was similar at the experimental endpoint. Medium was not refreshed in order not to disturb the hypoxic environment. O_2_ concentration in the incubator was frequently monitored. HIF1α nuclear stabilization and CA9 induction were measured to verify hypoxia.

Primary antibodies (Abs) included those targeting Carbonic Anhydrase IX (NB100-417SS; Novus, Englewood, CO, USA), p-YAP/p-TAZ (#13008; Cell Signaling Technology, Danvers, MA, USA), YAP (#4912; Cell Signaling Technology), TAZ (#83669; Cell Signaling Technology), Src (#2108; Cell Signaling Technology), p-Src (#6943; Cell Signaling Technology), HIF1α (#610959; BD Biosciences, Franklin Lakes, NJ, USA), GAPDH (sc-32233; Santa Cruz, Dallas, TX, USA), β-actin (sc-47778; Santa Cruz), Tubulin (T-9026; Sigma-Aldrich, Burlington, MA, USA). For immune fluorescence microscopy, secondary Abs were Alexa 488–linked anti-mouse IgG (Invitrogen, Waltham, MA, USA) and Alexa 546–linked anti-rabbit IgG (Invitrogen). For Western blotting, secondary Abs included horseradish peroxidase (HRP)–linked anti-rabbit or mouse IgG (Jackson ImmunoResearch Laboratories Inc., West Grove, PA, USA).

Hoechst was purchased from Thermo Fisher, Waltham, MA, USA. BV02, Verteporfin and DMOG were purchased from Sigma-Aldrich. PP2 and Dasatinib were purchased from Selleckchem, Planegg, Germany.

### 4.2. Sulforhodamine B (SRB) Assay

SRB assays were used to measure cell proliferation. 3000–5000 cells/well were seeded in 96-well plates. At indicated time points, 50% TCA was used to fix the plates and 0.4% SRB was added and incubated at RT avoiding light. Plates were washed with 1% acetic to remove unbound SRB and air dried. 10 mM Tris was added into wells to extract protein and absorbance was measured at 540 nm by plate reader (Tecan Infinite M1000, Männedorf, Switzerland).

### 4.3. Gene Silencing, Deletion, and Ectopic Expression

For transient siRNA mediated gene silencing 50 nM SMARTpool siGENOME siRNAs (Dharmacon, Lafayette, CO, USA) were transfected using the transfection reagent INTERFERin (Polyplus, Illkirch-Strasbourg, France) according to the manufacturer’s procedures. Controls included a SMARTpool targeting the luciferase gene, GAPDH, and a mixture of siRNAs from the whole genome library targeting all kinases in the human genome, diluted to the same total siRNA concentration as used for the SMARTpools (kinase pool).

Lentiviral supernatants were generated in HEK293 cells and used for transduction of target cells in combination with 5 µg/mL polybrene. For stable gene silencing, cells were transduced using lentiviral shRNA vectors (LentiExpress; Sigma-Aldrich, Saint Louis, MO, USA). Transduced cells were selected in medium containing puromycin. An shRNA targeting enhanced green fluorescent protein (eGFP) served as control.

For CRIPSR/Cas9 knockout, cells were transduced with lentiviral Edit-R Tre3G promotor-driven Cas9 (Dharmacon) and selected by blasticidin. Limited dilution was used to generate Cas9 monoclonal cells. Subsequently, Cas9-monoclonal cells were transduced with U6-gRNA: hPGK-puro-2A-tBFP containing control non-targeting or YAP targeting sgRNAs (Sigma) and bulk selected by puromycin. Knockout was induced by exposure to doxycycline for 48 h.

For ectopic expression of TAZ-S89A, a lenti-TAZ-S89A-IRES-GFP lentiviral construct was kindly provided by Gangyin Zhao and Ewa Snaar Jagalska, Leiden University, and verified by sequence analysis. Cells were transduced and bulk sorted by fluorescence-activated cell sorting (FACS; Sony, SH800S Cell Sorter, San Jose, CA, USA). Sorted cells were cultured for 2 days prior to use in experiments.

### 4.4. Western Blot, Cell Fractionation and Immunofluorescence Microscopy

For Western blot, cells were lysed with RIPA buffer containing 1% protease/phosphatase inhibitor cocktail (PIC, Sigma-Aldrich, P8340). Samples were separated by SDS–polyacrylamide gel electrophoresis and transferred to PVDF membranes (Millipore), incubated with primary Abs overnight at 4 °C followed by HRP-labelled secondary Abs (Jackson ImmunoResearch Laboratories Inc., West Grove, PA, USA) for 1 h at RT, and imaged with enhanced chemiluminescence substrate mixture (ECL Plus, Amersham, GE Healthcare, Chicago, IL, USA). Blots were imaged using an Amersham Imager (GE Healthcare Life Science, Chicago, IL, USA).

For cell fractionation, cells were lysed using the FractionPREP Cell Fractionation Kit (BioVision, Waltham, MA, USA) according to the manufacturer’s procedures.

For immunofluorescence microscopy, cells were fixed in 4% paraformaldehyde, permeabilized in 0.3% Triton X-100, blocked with 0.5% BSA for 30 min, and incubated with primary Abs overnight at 4 °C. The next day, cells were incubated with fluorescently labelled secondary Abs in combination with Hoechst33258 nuclear staining for 1 h under RT avoiding light. Images were taken with Nikon Eclipse Ti microscope and analyzed using the Intensity Ratio Nuclei Cytoplasm Tool (RRID:SCR_018573) in ImageJ (https://imagej.nih.gov/ij/). For quantitative image analysis ≥4 field of views were used per biological replicate. Scale bars shown in images represent 50 µm.

### 4.5. TempO-Seq and RT-qPCR

For TempO-Seq, 3000–5000 cells/well were plated in 96-well plates and incubated 5 days under normoxia or hypoxia after which the wells were washed once with cold PBS and lysed using TempO-Seq lysis buffer (Bioclavis, Glasgow, Scotland, United Kingdom) for 15 min at RT. Samples were stored at −80 °C before shipping to BioClavis for whole genome TempO-Seq analysis [63]. Expression data were shown as counts per probe. An in-house R script was used for count normalization and determining differential gene expression. The library size (total number of reads per sample) was set as 100,000 reads and samples below this size were removed. DESeq2 package was used to normalize counts and calculate differentially expressed genes (DEGs). DEGs were filtered by |log_2_Foldchange | > 1 and adjusted *p*-value (padj) < 0.05.

For RT-qPCR, total RNA was isolated by RNeasy Plus Mini Kit (Qiagen) and cDNA was synthesized by the RevertAid H Minus First Strand cDNA Synthesis Kit (Thermo Fisher Scientific, Waltham, MA, USA). Real-time qPCR was performed in triplicate using SYBR Green PCR (Applied Biosystems, Waltham, MA, USA) on QuantStudioTM 6 Flex Real-Time PCR system (Applied Biosystems). The following qPCR primer sets were used: β-actin forward (fw), 5′-ATTGCCGACAGGATGCAGAA-3′; β-actin reverse (rev), 5′-GCTGATCCACATCTGCTGGAA-3′; CDK3 forward (fw), 5′-TTCCTGGTCCACTTAGGGAAG-3′; CDK3 reverse (rev), 5′-CCAGCTCTTTCGTATCTTTCGT-3′.

CTGF forward (fw), 5′-GTTTGGCCCAGACCCAACTA-3′; CTGF reverse (rev), 5′- GGCTCTGCTTCTCTAGCCTG-3′; AXL forward (fw), 5′-CGTAACCTCCACCTGGTCTC-3′; AXL reverse (rev), 5′-TCCCATCGTCTGACAGCA-3′; CYR61 forward (fw), 5′-AAGAAACCCGGATTTGTGAG-3′; CYR61 reverse (rev), 5′-GCTGCATTTCTTGCCCTTT-3′. Relative mRNA expression was calculated after correction for the control (β-actin) using the 2^−ΔΔCT^ method.

### 4.6. RNA-Seq and TempO-Seq Data Analysis in Cell Lines and Clinical Samples

Known YAP/TAZ target genes were identified from existing literature [36,37]. TEAD1-4 binding genes were identified by transcription factor enrichment analysis using the DoRothEA tool version 2 (https://dorothea.opentargets.io/#/, accessed on 14 April, 2021) [64] with log2 normalized values as input as we have described previously [65]. In short, an average fold change was calculated over all probes for each gene and used to determine z-scores (hypoxia versus normoxia). The Viper package was used to determine transcription factor enrichment providing a normalized enrichment score per transcription factor [66]. For complete hierarchical clustering genes were uploaded to the OmicStudio tools at https://www.omicstudio.cn/tool (accessed on 13 June 2021).

RNA-seq data for a panel of 52 breast cancer lines [33] were used to identify DEGs in distinct breast cancer subtypes (basal A, basal B and luminal). RNA sequencing data from The Cancer Genome Atlas (TCGA) were obtained using the January 2017 version of TCGA Assembler R package [67]. The log2 normalized values were used for further analyzing. Data from solid primary tumor tissue samples were used [68]. Gene expression in different subtypes (ERpos and TNBC) was plotted in R v.3.6.3.

Breast Invasive Carcinoma mRNA expression from the same TCGA breast cancer study with metadata from basal-like, luminal A and luminal B breast cancer subtypes [69] was downloaded from cBioPortal [70]. Correlations of HIF1A with YAP1 and HIF1A with WWTR1 (NCBI Gene IDs: 3091, 10413 and 25937) were plotted using tidyverse, cowplot, and patchwork in R v.4.1.2.

Gene set enrichment analysis (GSEA) was performed using OmicStudio tools at https://www.omicstudio.cn/tool (accessed on 12 July 2020). Further enrichment analysis was performed using the Metascape platform (https://metascape.org, accessed on 10 May 2020) [71].

### 4.7. Kinase-Substrate Predictions of PP2 and Dasatinib

Activity values of human kinase proteins were collected from ChEMBL (version 27) [72] by filtering with taxon identifier and protein classification level 2 set to 9606 and “Kinases” respectively. Records with unassigned pChEMBL values and molecules with molecular weight greater than 800 Da were discarded. The dataset consisted in 311,249 compound-protein interactions from 124,307 compounds and 422 kinases. Molecules were represented with 67 Pipeline Pilot molecular descriptors (Appendix A) and kinase sequences with autocross-covariances [73] and domain averages [74] of BLOSUM [75], ProtFP [76], SSIA AM1 [77] and Z-scales [78] protein descriptors. Autocross-covariances were computed with a lag of 20 amino acids. For domain averages, protein sequences were split into 50 equal parts-where part length differed based on protein length. For each part the mean average value of each dimension was calculated, and the global mean average added, yielding 50 * d + 1 values, where d is the dimension of each protein descriptor (e.g., 50 * 10+1 for the BLOSUM descriptor). A training and holdout test set were derived using 70 and 30% of the data, respectively. An Extreme Gradient Boosting [79] (XGBoost version 1.4.2) regressor was fitted using 5-fold cross-validation with the following parameters: 100 maximum trees, learning rate of 0.3, max depth of 7, descriptor fraction of 0.7 and data fraction of 1.0. This proteochemometric model had an average cross-validated coefficient of determination (R^2^) of 0.66, Spearman rank correlation coefficient of 0.81 and root-mean-square error (RMSE) of 0.84. When evaluated on the holdout test set, the model had Pearson correlation of 0.68, Spearman r of 0.65 and RMSE of 0.99. Affinities were then derived for the 422 kinases for the structure of PP2 and Dasatinib using the model and sorted decreasingly (Appendix A).

### 4.8. Statistics

Data were analyzed with Student’s *t*-test or one-way ANOVA using GraphPad Prism 7 with the exception of statistical analysis of clinical samples where Wilcoxon signed-rank test was used as indicated in figure legends.

## Figures and Tables

**Figure 1 ijms-23-10119-f001:**
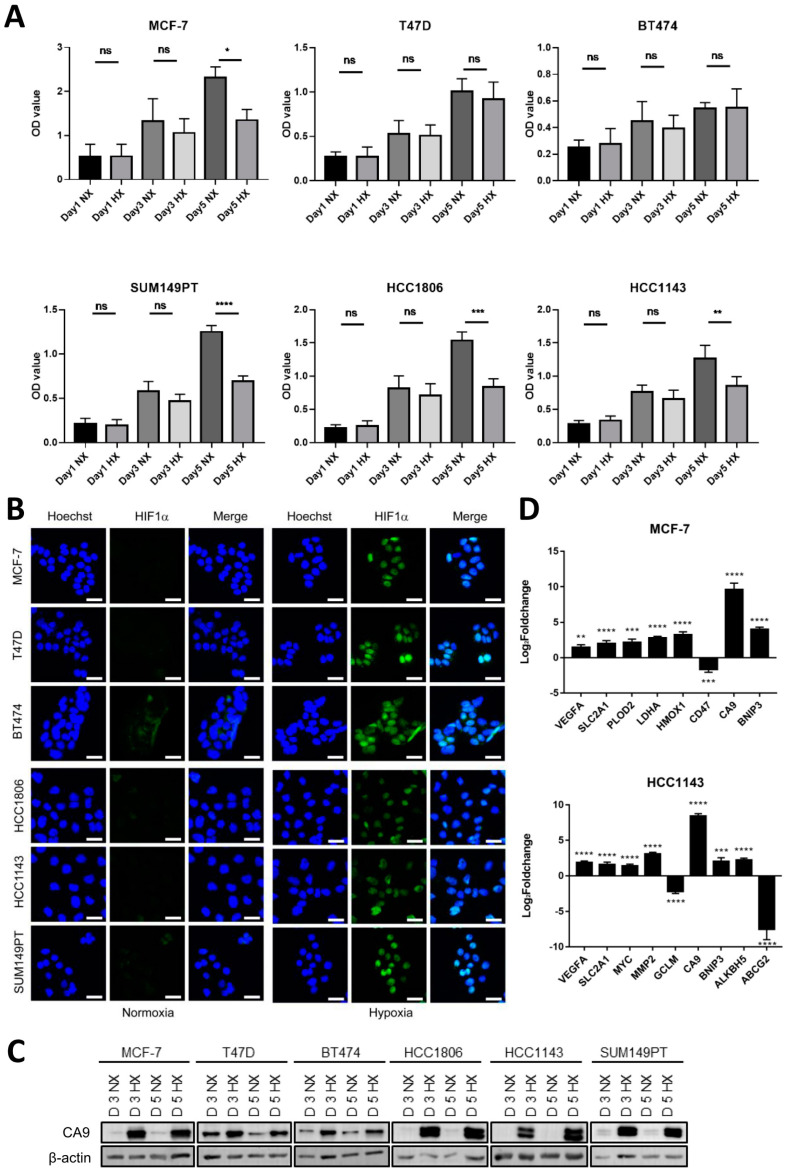
**Modulation of cell growth and activation of HIF signaling in response to hypoxia in a series of luminal and basal A cell lines.** (**A**) Cell growth analyzed by SRB for three luminal (MCF7, T47D and BT474) and three basal A (SUM149PT, HCC1806 and HCC1143) breast cancer cell lines grown under normoxia (21% O_2_; NX) or hypoxia (1% O_2_; HX) for 1, 3 or 5 days. Mean and SD of OD values from three biological replicates performed in triplicate are shown. ns, non-significant; *, *p* < 0.05; **, *p* < 0.01; ***, *p* < 0.001; ****, *p* < 0.0001, ns, not significant. (**B**) Luminal and basal A cell lines incubated under normoxia or hypoxia for 3 days analyzed for HIF1α expression and localization by confocal immunofluorescence microscopy. Blue, Hoechst; Green, HIF1α Ab. One representative experiment of three biological replicates is shown. (**C**) CA9 expression analyzed by Western blot after 3 and 5−day incubation under normoxia or hypoxia for the indicated luminal and basal A cell lines. (**D**) Identification of known HIF1 responsive genes [31] in TempO-Seq data comparing 5−day incubation under normoxia or hypoxia in MCF7 and HCC1143 cells. Mean Log_2_foldchange for hypoxia relative to normoxia and SD of triplicate measurements is shown. **, padj < 0.01; ***, padj < 0.001; ****, padj < 0.0001.

**Figure 2 ijms-23-10119-f002:**
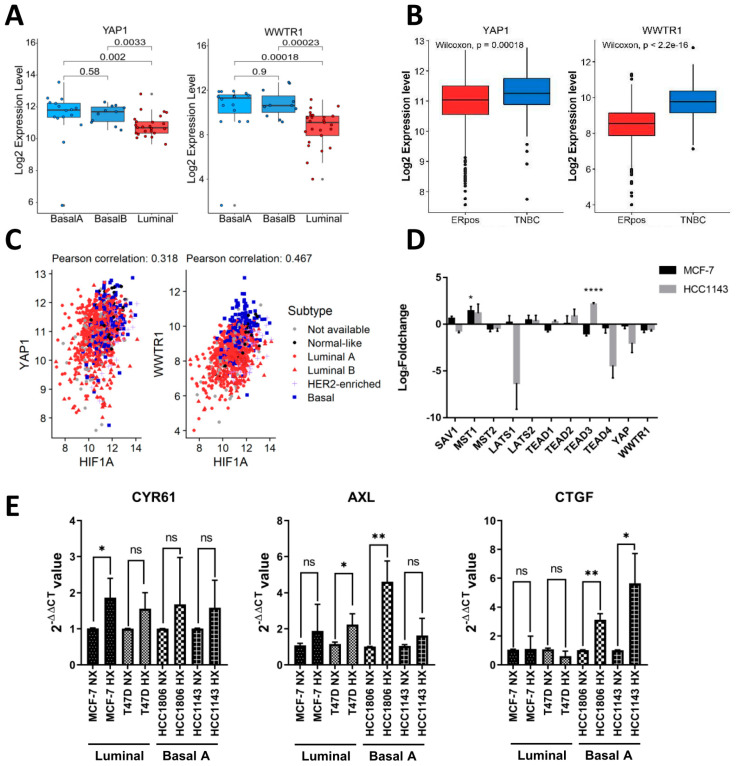
**Expression of YAP and TAZ in breast cancer cell lines and tumors.** (**A**) RNA expression level of YAP and TAZ (WWTR1) from RNA-Seq data for 52 human breast cancer cell lines classified by luminal−, basal A−, or basal B−like subtype [33]. p-value calculated using One−way ANOVA. (**B**) Log_2_RNA expression levels of YAP and TAZ in ER positive and TNBC clinical samples. p-value calculated using Wilcoxon signed-rank test. (**C**) Correlation between RNA expression levels for HIF1α and YAP and for HIFα and TAZ determined using cBioPortal data from basal-like, HER2 enriched, luminal A and luminal B breast cancer subtypes. (**D**) Expression of YAP, TAZ, TEAD1−4, and upstream kinases in the Hippo signaling cascade determined in TempO-Seq data. Log_2_foldchange under hypoxia relative to normoxia is shown in MCF7 and HCC1143. Mean and SD of triplicate measurements is shown. *, padj < 0.05; ****, padj < 0.0001. (**E**) qPCR experiment showing expression of the indicated YAP/TAZ targets genes in the indicated luminal and basal A TNBC cell lines exposed to normoxia or hypoxia after 5 days. Mean expression relative to β−actin is shown. Error bars indicate SD for triplicate measurements. *, *p* < 0.05; **, *p* < 0.01; ns, not significant.

**Figure 3 ijms-23-10119-f003:**
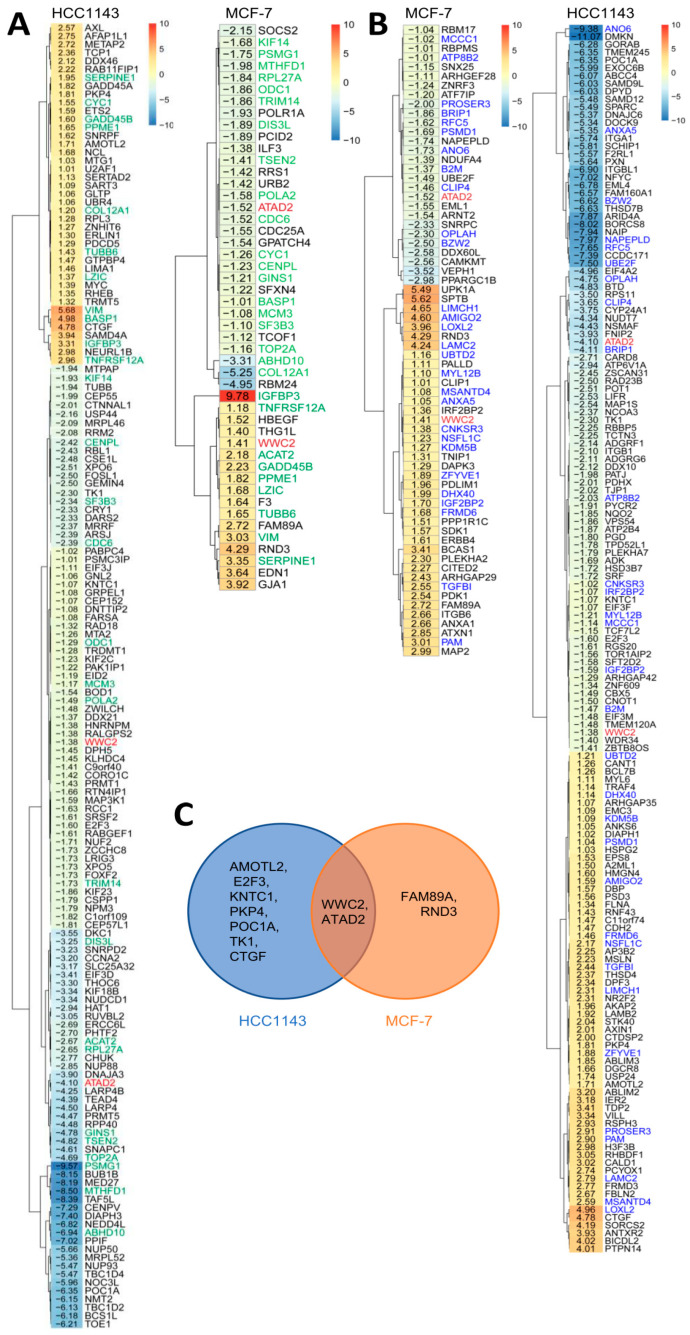
**Identification of candidate YAP/TAZ/TEAD target genes in hypoxia-responsive genes in MCF7 and HCC1143 cells.** (**A**,**B**) Heatmaps showing intersection of TempO-Seq data for hypoxia-regulated genes in MCF7 and HCC1143 cells with YAP/TAZ/TEAD target genes reported in literature (**A**; [36,37]) and candidate TEAD1-4-binding genes identified using DoRothEA (**B**; [38]). Values are Log_2_ fold change under hypoxia relative to normoxia. Genes labeled green in (**A**) or labeled blue in (**B**) are shared between MCF7 and HCC1143 in the respective analyses. Genes labeled red in (**A**) and (**B**) are shared in both analyses for both cell lines. (**C**) Venn diagram showing distinct and overlapping hypoxia regulated genes for MCF7 and HCC1143 identified both in literature and by DoRothEA.

**Figure 4 ijms-23-10119-f004:**
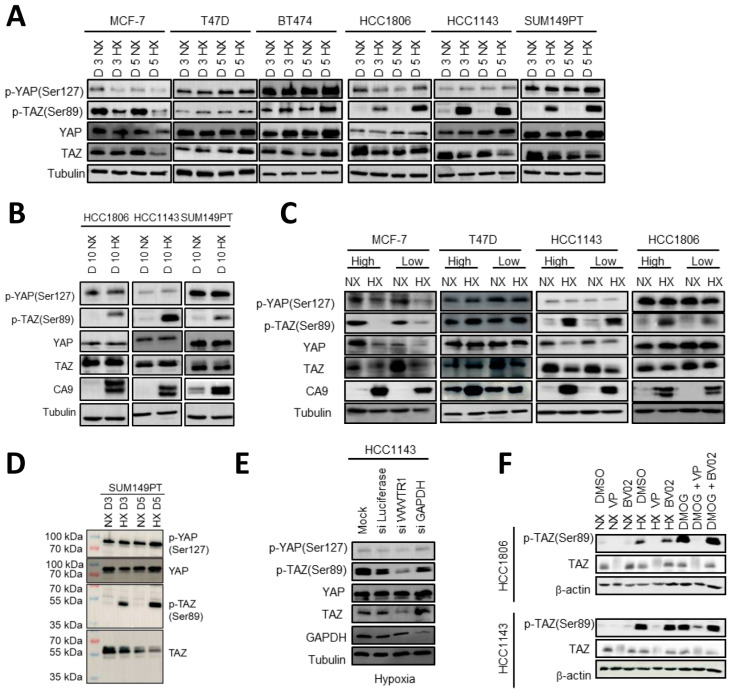
**HIF1-mediated phosphorylation of TAZ at Ser89 under hypoxia in****basal A but not luminal breast cancer cells.** (**A**,**B**) Western blot showing YAP, TAZ, p-YAP (Ser127) and p-TAZ (Ser89) for three luminal (MCF7, T47D and BT474) and three basal A (SUM149PT, HCC1806 and HCC1143) breast cancer cell lines incubated under normoxia (NX) or hypoxia (HX) for 3 or 5 (**A**) or 10 days (**B**). Tubulin serves as loading control. (**C**) Western blot showing the indicated (phospho-)proteins for the indicated luminal and basal A cells seeded at low density (6 × 10^4^ cells/petri dish; subconfluent) or high density (15^e^4 cells/petri dish; confluent) and exposed for 5 days to normoxia or hypoxia. (**D**) Western blot analysis of the indicated (phospho-)proteins for SUM149PT exposed for 3 or 5 days to normoxia or hypoxia. (**E**) Western blot analysis of the indicated (phospho-)proteins for HCC1143 cells treated with TAZ (WWTR1) SMARTpool siRNAs or the indicated control SMARTpool siRNAs and exposed for 5 days to hypoxia. (**F**) Western blot showing p-TAZ (Ser89) and TAZ in HCC1806 and HCC1143 cells cultured in absence or presence of Verteporfin (5 µM) or BV02 (5 µM) while exposed to normoxia, hypoxia, or DMOG (1 mM) for 3 days. β-actin serves as loading control.

**Figure 5 ijms-23-10119-f005:**
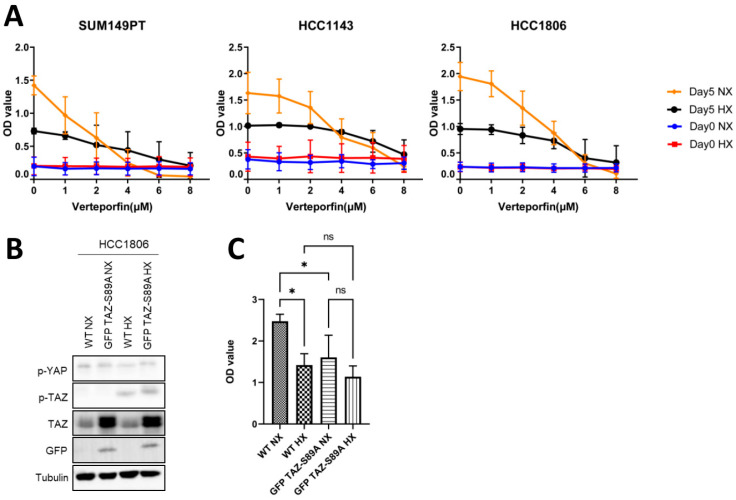
**Effect of YAP/TAZ/TEAD complex inhibitor, verteporfin and expression of GFP TAZ-S89A in basal A cells.** (**A**) Results of SRB assay for three basal A breast cancer cell lines at day 0 (indicating equal cell seeding densities) and after 5 days incubation under normoxia or hypoxia in presence of 0 (DMSO vehicle control), 1, 2, 4, 6 or 8 µM Verteporfin. Mean of raw OD values and SD for three biological replicates is shown. (**B**,**C**) GFP TAZ-S89A HCC1806 cells and wildtype HCC1806 cells (WT) were incubated under normoxia and hypoxia for 5 days and analyzed by Western blot (**B**) or SRB assay (**C**). For SRB assays, mean and SD of OD values from three biological replicates performed in triplicate are shown. ns, non-significant; *, *p* < 0.05; ns, not significant.

**Figure 6 ijms-23-10119-f006:**
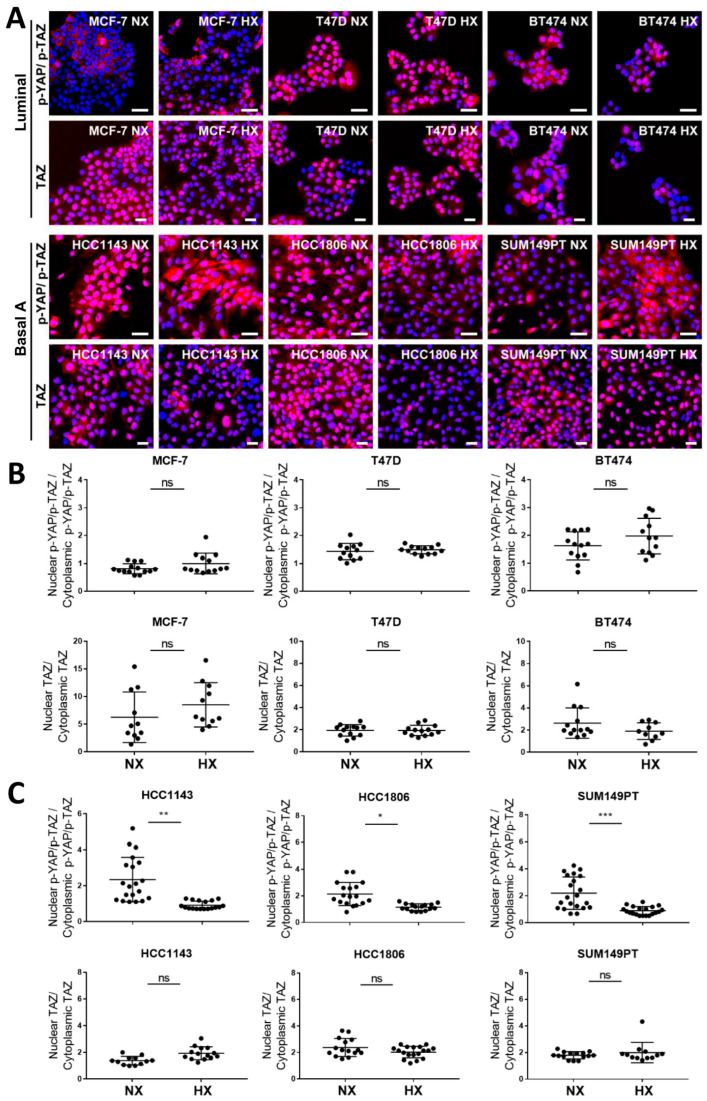
**Localization of TAZ and phosphorylated YAP/TAZ in luminal breast cancer and basal A cells under normoxia and hypoxia.** (**A**) Localization of TAZ and p-YAP/TAZ (using an Ab detecting both p-YAP(Ser127) and p-TAZ (Ser89)) determined by confocal immune fluorescence microscopy in luminal (MCF7, T47D and BT474) and basal A (SUM149PT, HCC1806 and HCC1143) breast cancer cell lines cultured under normoxia (NX) or hypoxia (HX) for 5 days. Blue, Hoechst; Red, Abs. (**B**,**C**) Quantification of nuclear/cytoplasmic distribution of TAZ and p-YAP/TAZ in luminal (**B**) and basal A cells (**C**) cultured under normoxia or hypoxia, calculated form images in (**A**). Y-axis indicates percentage of the total intensity in the nuclei area versus percentage of the total intensity in the cytoplasm area. *, *p* < 0.05; **, *p* < 0.01; ***, *p* < 0.001; ns, not significant.

**Figure 7 ijms-23-10119-f007:**
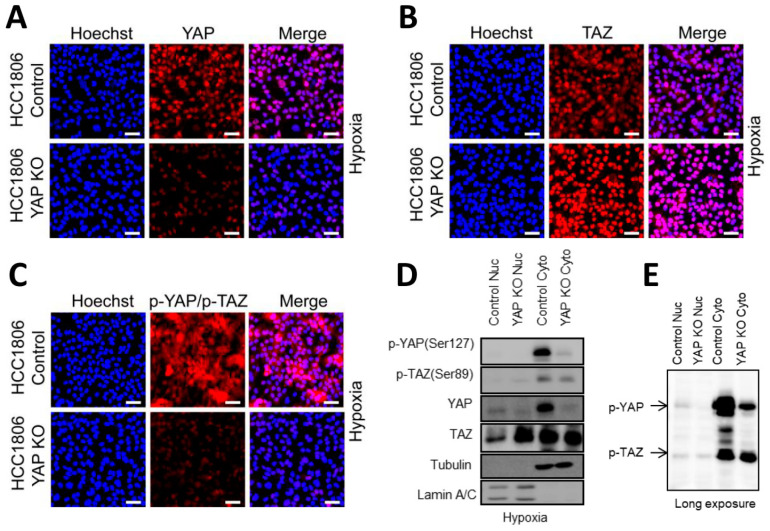
**Subcellular localization of phospho-YAP/TAZ upon YAP knockout in HCC1806 cells under hypoxia.** (**A**–**C**) YAP (**A**), TAZ (**B**) and phospho-YAP/TAZ (**C**) expression and localization analyzed by confocal immunofluorescence microscopy after 5-day incubation under hypoxia in CAS9 expressing HCC1806 cells transduced with YAP- or non-targeting control lentiviral sgRNA constructs. (**D**) Detection of YAP, TAZ, p-YAP (Ser127), and p-TAZ (Ser89) by Western blot after 5-day incubation under hypoxia in nuclear and cytoplasmic fractions of CAS9 expressing HCC1806 cells transduced with YAP- or non-targeting control lentiviral sgRNA constructs. Tubulin and Lamin A/C serve as loading controls for cytoplasmic and nuclear fraction, respectively (**E**) Longer exposure of Western blot used for (**D**).

**Figure 8 ijms-23-10119-f008:**
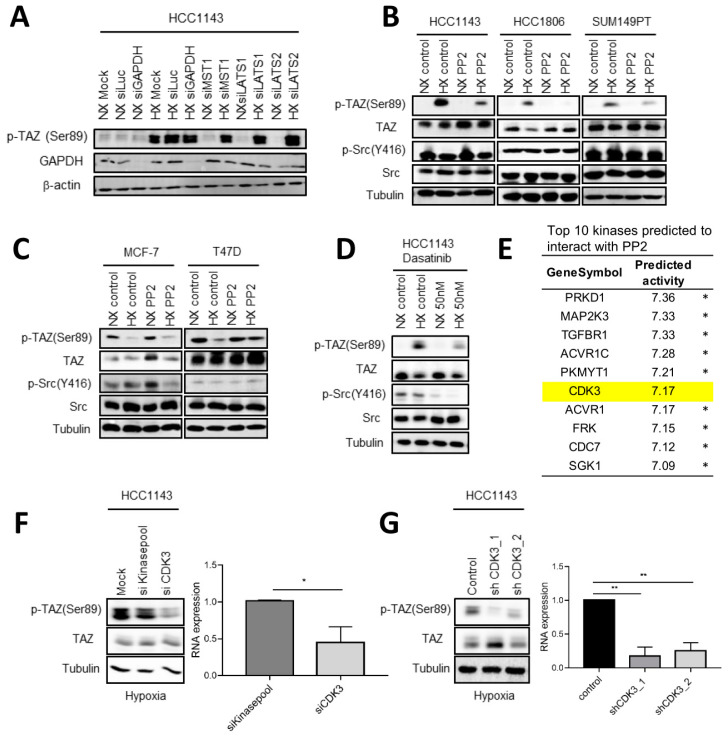
**Src and CDK3 mediate phosphorylation of TAZ on Ser89 under hypoxia in basal A cells.** (**A**) Western blot analysis of p-TAZ (Ser89) in HCC1143 cells treated with the indicated SMARTpool siRNAs targeting components of the Hippo signaling cascade or luciferase or GAPDH as controls and exposed for 5 days to normoxia (NX) or hypoxia (HX) from two biological replicates. β-actin serves as loading control. (**B**,**C**) Western blot analysis of the indicated (phospho-)proteins in the indicated basal A (**B**) or luminal breast cancer cell lines (**C**) exposed to normoxia or hypoxia for 3 days in absence or presence of 1µM PP2. Tubulin serves as loading control. (**D**) Western blot analysis of the indicated (phospho-)proteins in HCC1143 basal A cells exposed to normoxia or hypoxia for 3 days in absence or presence of 50nM Dasatinib. (**E**) Top 10 kinases identified as candidate PP2 substrates by ChEMBL ranked by activity. * Indicates kinases where interaction with Dasatinib scored higher interaction with PP2. (**F**,**G**) Western blot analysis of TAZ and p-TAZ (Ser89) in HCC1143 cells treated with control (kinase pool) or CDK3 targeting SMARTpool siRNAs (**F**) or in HCC1143 cells transduced with control non-targeting or two different CDK3 targeting lentiviral shRNA constructs (**G**). Cells were exposed for 5 days to hypoxia. Graphs show qPCR analysis of gene silencing efficacy of the CDK3 targeting siRNA SMARTpool (**F**) or the CDK3 targeting lentiviral shRNA constructs (**G**). Mean expression relative to β-actin is shown. Error bars indicate SD for triplicate measurements. *, *p* < 0.05; **, *p* < 0.01.

## Data Availability

TempO-seq data supporting the results of this article will be available in the BioStudies database from EMBL-EBI.

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
