# Peer review of "Hypoxia Triggers TAZ Phosphorylation in Basal A Triple Negative Breast Cancer Cells"

_ijms, 2022, doi:10.3390/ijms231710119_

Round 1

Reviewer 1 Report

The paper titled “Hypoxia triggers TAZ phosphorylation in basal A triple negative breast cancer

cells” aims to identify the differences in luminal and basal A breast cancer cells in response to
hypoxic tumor micro-environment in the context of YAP/TAZ/TEAD signaling. Towards this
end, the authors have compared luminal and basal A/B breast cancer cells in normoxic versus
hypoxic conditions and delineated shared and unique transcriptional signatures. Using a
variety of expression models and in vitro experiments, the authors conclude that hypoxia
mediated phosphorylation of TAZ in basal A cells involves Src and CDK3 kinases. Overall, the
experiments are well designed, and the results fully support the conclusion of the paper. I
would appreciate if the authors could respond to the following minor comments:

1. Cell culture conditions: Please elaborate on the incubator and culture conditions used
for hypoxia. In my experience, in hypoxic conditions (1% oxygen), the media turns
acidic after 72 h. For experiments extending for 5-10 days:

(a) Was there a need to change culture media?

(b) In hypoxic conditions, a subset of cells undergo cell death as well. Did you
observe higher cell death in hypoxia when compared to normoxia?

2. CD47 is known to be up-regulated in hypoxic conditions in breast cancer cells (including
MCF-7) (Zhang et al., PNAS, 2015). How would you explain the down-regulation of
CD47 in your results (Fig. 1D)?

3. Microscopy:

(a) For quantifying the ratio of nuclear and cytoplasmic distribution of TAZ, how many
frames were captured per biological replicate?

(b) Isn’t separating nuclear and cytoplasmic protein fractions, followed by western blot
analysis a better way of quantifying protein distribution than immunofluorescence
microscopy?

(c) In Figure S5, staining with HIF1 antibody does not appear to be necessary for YAP
quantification.

4. Western blots:

(a) Figure 1C full blot: What are the last 2 lanes after D10HX? What are the bands on
top (~100kDa) in BT474?

(b) In full blot (page 2), is the Label Figure 1C, 4B correct? (Figure 4A, 4B is shown in
page 4)

(c) In the full blot, many labels appear on top the bands. Please make sure the labels
dont overlap with the bands.

(d) p-TAZ bands: Which antibody was used for p-TAZ? I think it is not mentioned in
Materials and Methods. How would you explain the smaller size of p-TAZ bands?

5. Fig 2E: How many days/hours of hypoxia was given for qPCR of CYR61, AXL, CTGF?

6. Fig 4A: The authors note that total TAZ protein level was not affected by hypoxia.
However, for MCF-7 and HCC1143, the total TAZ signal for 5D HX appears to be much
weaker than other 3 samples.

7. PP2 induces its effects within hours. Is it known to exert its affect even up to 72 hours?

Author Response

We thank the reviewers for their detailed analysis and positive evaluation of our manuscript. Please find below our answers to their comments:

The paper titled “Hypoxia triggers TAZ phosphorylation in basal A triple negative breast cancer cells” aims to identify the differences in luminal and basal A breast cancer cells in response to hypoxic tumor micro-environment in the context of YAP/TAZ/TEAD signaling. Towards this end, the authors have compared luminal and basal A/B breast cancer cells in normoxic versus hypoxic conditions and delineated shared and unique transcriptional signatures. Using a variety of expression models and in vitro experiments, the authors conclude that hypoxia mediated phosphorylation of TAZ in basal A cells involves Src and CDK3 kinases. Overall, the experiments are well designed, and the results fully support the conclusion of the paper. I would appreciate if the authors could respond to the following minor comments:
1. Cell culture conditions: Please elaborate on the incubator and culture conditions used
for hypoxia. In my experience, in hypoxic conditions (1% oxygen), the media turns
acidic after 72 h. For experiments extending for 5-10 days:
(a) Was there a need to change culture media?

We did not change medium in order not to disturb the hypoxic environment. O2 concentration in the incubator was regularly monitored. HIF1α stabilization and CA9 induction were additionally measured to verify hypoxia. For cells cultured under long-term hypoxia, we adjusted initial seeding cell densities for different cell lines to make sure the final cell confluency was similar after long-term culture.

This information was added to the “experimental procedures” section (4.1) in the revised manuscript.

(b) In hypoxic conditions, a subset of cells undergo cell death as well. Did you
observe higher cell death in hypoxia when compared to normoxia?

No, we did not observe any signs of loss of cell viability triggered by hypoxia.

  1. CD47 is known to be up-regulated in hypoxic conditions in breast cancer cells (including
    MCF-7) (Zhang et al., PNAS, 2015). How would you explain the down-regulation of
    CD47 in your results (Fig. 1D)?

Fig1D shows the results of targeted RNA-seq but not qPCR. Notably, the results are shown for 5-days culture under hypoxia. It is well known that effects of acute versus chronic hypoxia can be markedly different. The qPCR in the paper of Zhang et al. detected CD47 expression after less than 24hrs hypoxic exposure. Therefore, it is difficult to directly compare these results.

Parenthetically, we have checked another dataset where we compared short and long exposure to hypoxia using the same targeted RNA-seq approach. Here, we did not observe a significant change in CD47 mRNA expression either up or down at short exposure to hypoxia. This data will be part of another manuscript. 

  1. Microscopy:
    (a) For quantifying the ratio of nuclear and cytoplasmic distribution of TAZ, how many
    frames were captured per biological replicate?

For this quantification ³4 different field of views were used per biological replicate. This information has been added to the “experimental procedures” section (4.4) in the revised manuscript.

(b) Isn’t separating nuclear and cytoplasmic protein fractions, followed by western blot
analysis a better way of quantifying protein distribution than immunofluorescence
microscopy?

We use quantitative image analysis to retrieve information on subcellular localization from confocal microscopy images. This is an accepted approach suitable for larger numbers of conditions/cell lines. Indeed, as the reviewer suggests, for our key finding that the shift of p-TAZ under hypoxia to the cytoplasm, we use separating nuclear and cytoplasmic protein fractions followed by Western blot analysis as an independent second approach for one of the basal cell lines (Fig 7).

(c) In Figure S5, staining with HIF1 antibody does not appear to be necessary for YAP
quantification.

Indeed, the HIF signal was not required for YAP quantification, but we used it to verify that the cells were in a hypoxic condition.

  1. Western blots:
    (a) Figure 1C full blot: What are the last 2 lanes after D10HX? What are the bands on
    top (~100kDa) in BT474?

Yes. The last 2 lanes are D15 NX and D15 HX. The information has now been added in the supplemental full blot as requested.

For the ~100kDa band in the BT474 blot we do not know. The blot was not exposed to another Ab. This likely reflects a specific band. The molecular weight of CA9 is ~55kDa and this is what we show in the manuscript.

(b) In full blot (page 2), is the Label Figure 1C, 4B correct? (Figure 4A, 4B is shown in
page 4)

Indeed, this is correct. In full blot (page 2), it shows CA9 expression after 3, 5, and 10- day exposure under normoxia and hypoxia. Fig1C includes D3 and D5. Fig 4B includes D10.

In full blot (page 4), it shows YAP, TAZ, p-YAP and p-TAZ expression after 3, 5, and 10- day exposure under normoxia and hypoxia. Fig4A includes D3 and D5. Fig 4B includes D10.

(c) In the full blot, many labels appear on top the bands. Please make sure the labels
don’t overlap with the bands.

As requested, this has been corrected.

(d) p-TAZ bands: Which antibody was used for p-TAZ? I think it is not mentioned in
Materials and Methods. How would you explain the smaller size of p-TAZ bands?

We used the phospho-YAP/TAZ Ab from Cell Signaling Technology (#13008). This information was provided in the experimental procedures (chapter 4.1).

The mechanism for the reduced size of p-TAZ as compared to TAZ is not fully understood. TAZ pre-mRNA splicing variation has been described, which may be affected by hypoxia. Alternatively, the lower size may be related to changes in post translational modification such as glycosylation or ubiquitination. To confirm that the band does reflect TAZ we show siWWTR1 leads to a loss of this p-TAZ signal. Moreover, Verteporfin, which lead to loss of TAZ signal leads to a concomitant loss of the p-TAZ signal.

  1. Fig 2E: How many days/hours of hypoxia was given for qPCR of CYR61, AXL, CTGF?

Exposure to hypoxia was 5 days. This has been added in the figure legend Line159 in the revised manuscript.

  1. Fig 4A: The authors note that total TAZ protein level was not affected by hypoxia.
    However, for MCF-7 and HCC1143, the total TAZ signal for 5D HX appears to be much
    weaker than other 3 samples.

Indeed, as the reviewer correctly notices there is some variation. We concluded that overall, TAZ expression in most of cell lines was not affected. I.e., there was no trend that was conserved for all cell lines or all cell lines of one subtype. We have now modified the description of this data to avoid confusion in Line199 in the revised manuscript.

  1. PP2 induces its effects within hours. Is it known to exert its affect even up to 72 hours?

Yes, there are other studies using PP2 effectively in experiments lasting up to 72hrs. Please see for example, PMID: 22570868 (e.g., Fig 1); PMID: 28671964 (e.g., Fig 3, 4).

Reviewer 2 Report

General comments

- The article is very interesting 

- The methods and results  are  well explained

- A lots of assays was performed

Minor comments

- The discussion is a little bit narrow and not sufficient to explain the results

- One or two sentence conclusion makes more sound for the article

Specific comments

1. Figure 8B, there is a difference in the expression of p-TAZ, TAZ, P-src, indicates there is a cell-type specific difference with in the basal type A.,  which needs to be discussed in the discussion part

2. Line 411 and 457 the number of cells seeded is not mentioned

Author Response

We thank the reviewers for their detailed analysis and positive evaluation of our manuscript. Please find below our answers to their comments:

General comments

- The article is very interesting 

- The methods and results are well explained

- A lots of assays was performed

Minor comments

- The discussion is a little bit narrow and not sufficient to explain the results

- One or two sentence conclusion makes more sound for the article

We have re-read the discussion and believe we have discussed the key findings of our study in context as well as brought up key points that require further study. As suggested by the reviewer, we have added short concluding sentences at the end of the discussion section in the revised manuscript.

Specific comments

  1. Figure 8B, there is a difference in the expression of p-TAZ, TAZ, p-Src, indicates there is a cell-type specific difference within the basal type A., which needs to be discussed in the discussion part

The 3 Western blot panels for the 3 cell lines in Figure 8B cannot be quantitatively compared. The signals in the 4 lanes within each panel can be quantitatively compared. We therefore cannot conclude as the reviewer suggests that there are “cell-type specific difference within the basal type A”. There will certainly be small differences in the amount of signal between different biological replicates, but each replicate shows the exact same pattern. Therefore, what we describe in the results and what can be indeed concluded is that for all 3 cell lines under hypoxia i) p-TAZ is induced under hypoxia, ii) total TAZ levels is in general not affected, and iii) there is no evidence for a loss of p-Src (Y416) in the cells treated with PP2.

  1. Line 411 and 457 the number of cells seeded is not mentioned

As requested, this has been corrected in the revised manuscript.